# Evaluation of Fungicides and Fungicide Application Methods to Manage Phytophthora Blight of Pigeonpea

**Mamta Sharma \***[ID]**, Ramanagouda Gaviyappanavar and Avijit Tarafdar**

International Crops Research Institute for the Semi-Arid Tropics (ICRISAT), Patancheru 502 324, India
* Correspondence: mamta.sharma@icrisat.org

**Abstract:** Phytophthora, a blight of pigeonpea caused by Phytophthora cajani, has been significantly increasing in major pigeonpea production regions of India. Limited information on infection with this pathogen and its epidemiology, as well as a lack of adequate resistant cultivars, is hampering the management of Phytophthora blight significantly. Therefore, five fungicides, viz., metiram + dimethomorph, cymoxanil + mancozeb, famoxadone + cymoxanil, mancozeb, and metalaxyl-M + mancozeb, were evaluated against *P. cajani* under control conditions to control zoospore induction, as well as the infection of zoospores, at the seedling stage. The half-maximal effective concentration (EC50) of fungicides for mycelial inhibition was calculated. The lowest EC50 was recorded in metiram + dimethomorph (0.17 μg/mL), followed by the metalaxyl-M + mancozeb (2.49 μg/mL) and cymoxanil + mancozeb (8.23 μg/mL) fungicides. The formation of the sporangium and zoospores was most significantly affected by metalaxyl-M + mancozeb, followed by metiram + dimethomorph and cymoxanil + mancozeb, in terms of sporangia viability and zoospore germination and encystment. Further, under glasshouse conditions, different fungicide application methods (e.g., seed-treatment; soil-drench; foliar-spray, either singly or in combinations) were evaluated with fungicides on susceptible (ICP 7119) moderately resistant pigeonpea (ICPL 99010, ICPL 20135 and ICPL 99048) cultivars. The seed-treatment + soil-drench, soil-drench + foliar-spray, and soil-drench of fungicide application methods were found to be effective in controlling the Phytophthora blight, at $p < 0.001$. A combination of the seed-treatment + soil-drench and soil-drench + foliar-spray methods, using metalaxyl-M + mancozeb or metiram + dimethomorph fungicides on moderately resistant cultivars (ICPL 99010), has a synergistic effect on the ability to control the Phytophthora blight at the seedling stage.

**Keywords:** pigeonpea; *Phytophthora cajani*; sporangia; zoospores; fungicides





## 1. Introduction

The pigeonpea (*Cajanus cajan* L.) is a perennial legume, widely cultivated as an annual crop in tropical and semitropical regions of South and Southeast Asia, Africa, and Latin America. It is highly nutritious, containing high levels of protein and several important free amino acids, e.g., tryptophan, methionine, and lysine. An estimated 4.49 million tons of pigeonpea are produced per year, and 63% of the total production occurs in India. In India, pigeonpea is cultivated in a 3.9-million-hectare area; this accounts for 72% of the total area used for pigeonpea cultivation across the world (FAOSTAT, 2021).

Phytophthora blight caused by *Phytophthora cajani* is one of the most important emerging pigeonpea diseases, after Fusarium wilt and sterility mosaic disease [1]. The disease was first observed in India [2] and subsequently addressed in other parts of the world [3,4]. The disease was initially classified as *Phytophthora drechsleri* f. sp. *cajani*, but later, it was amended to *Phytophthora cajani* [5]. Considering its distribution and severity, Phytophthora blight is an emerging threat to pigeonpea production across the world [6].

Under favorable conditions, *P. cajani* can infect pigeonpea plants at any growth stage and produce a variety of symptoms, including stem blight, stem rot, stem canker, and leaf

blight [1,7,8]. However, the disease was seen as minor until the year 2005–2006. After that, reoccurrence of this disease was reported often in many pigeonpea-growing regions in India [7–10]. The disease was reported to be more severe during the early crop growth stage, especially coupled with intermittent rains from June to September [11]. The disease affects serious proportions of crops in areas where excessive rainfall occurs within a short span of time followed by cloudy and hot/humid weather, the type of weather that persists during cropping seasons [12,13]. The farmer field survey report showed a complete loss of pigeonpea crops due to Phytophthora blight under favorable weather conditions [6].

*P. cajani*, a water mold or oomycete, is characterized by its restricted host range to members of *Cajanus* spp. [14]. The pathogen is both soil- and water-borne, but not seed-borne, in nature. Splashes from rain and the wind are the best-known contributors to dispersing zoospores across short distances. The pathogen can survive within infected crop debris for a year, and in the soil for a few years, even in the absence of a living host [1]. Management of the Phytophthora blight was achieved by employing different approaches, including resistant cultivars and cultural, chemical, and biological methods [15]. However, most of these reports pertaining to Phytophthora blight management practices are almost two decades old.

The recent devastating outbreak of the disease imposed a risk on pigeonpea production in many parts of India, and highlighted the need for robust research to understand the reoccurrence of the disease under climate change. The available resistant sources identified previously in [16–19] have now been proven to be susceptible to *P. cajani* under natural epiphytotic conditions [1,7–9]. Owing to the lack of reliable resistant sources against this pathogen [1,20], other options related to agronomic management practices might be helpful in reducing the disease incidence [21]. Chemical control methods, such as applying the fungicide Brestan-60 at the seedling stage and dressing seeds with metalaxyl [11], have been reported with varying degrees of efficacy.

Andrieu et al. [22] reported that famoxadone inhibited the growth of *P. infestans* during the different stages of its life cycle by causing lysis of sporangia and zoospores at the time of release and differentiation. Similar systematic studies on the efficacy of fungicides on *P. cajani* are limited, and very few studies have been reported on the effectiveness of fungicides in managing the Phytophthora blight in pigeonpea [23–25]. To the best of our knowledge, none of the studies has reported the efficacy of fungicides in the inhibition of sporangia and zoospore formation, discharge of zoospores, or infection in pigeonpea seedlings. In this study, we sought to (i) evaluate the efficacy of new fungicides on the mycelial growth and sporangia and zoospore induction in *P. cajani* at the host-independent stage; (ii) identify the relative responses of different fungicides and their application methods to Phytophthora seedling blight in pigeonpea plants under controlled conditions.

## 2. Materials and Methods

### 2.1. Phytophthora cajani Isolate and Its Cultural Conditions

A *P. cajani* isolate, ICPC 1 (KJ010534), was collected from the Legume Pathology repository (ICRISAT, Patancheru, India). This isolate was previously collected from pigeonpea fields in ICRISAT, Patancheru (GPS coordinates: 17.0911, 78.9581). Plants exhibiting typical Phytophthora blight lesions on their stems were selected for isolation of the pathogen. The isolate was sub-cultured on a 20% tomato extract agar (tomato extract, 200 mL; $CaCO_3$ (HiMedia Laboratories, Maharashtra, India), 2 g L$^{-1}$; and agar powder (HiMedia Laboratories), 20 g L$^{-1}$) and incubated (Percival Scientific's, Perry, IA, USA) at 25 $\pm$ 1 °C with a 12 h light/dark photoperiod (Figure 1). The pathogenicity of the isolate was determined by inoculating zoospores (1.5 $\times$ 105/mL) in a susceptible pigeonpea ICP 7119 cultivar under glasshouse conditions.

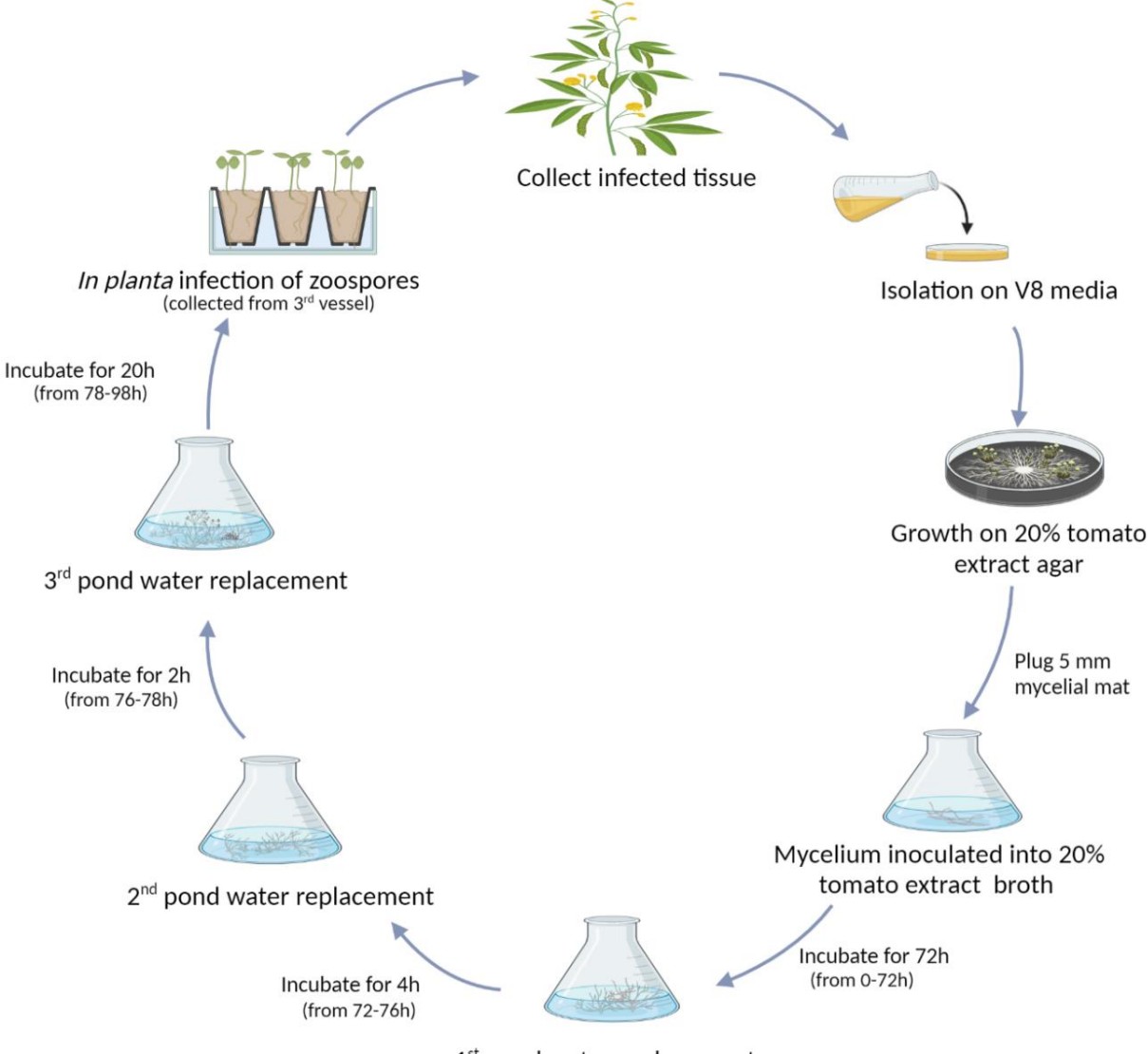

**Figure 1.** Schematic diagram of the protocol for production of sporangia and zoospores and in planta infection of *P. cajani* in pigeonpea seedlings. Sterilized pond water was used in each water replacement step. The time period was calculated from '0 h' (i.e., first inoculation of *P. cajani* in 20% tomato extract broth).

*2.2. Fungicides and Cultivars*

Five commercial fungicides, Acrobat (Metiram 44% + Dimethomorph 9% WG), Curzate (Cymoxanil 8% + Mancozeb 64% WP), Equation Pro (Famoxadone 16.6% + Cymoxanil 22.1% EC), Indofil M-45 (mancozeb 75% WP), and RidomilGold (Metalaxyl-M 4% + Mancozeb 64% WP) were selected based on their active ingredients, mode of action, target site, and efficacy on different Phytophthora species (Table 1). Stock solutions of these fungicides were prepared in sterile deionized water at different concentrations (Supplementary Table S1). Furthermore, the efficacy of the fungicides and their different application methods were assessed on *P. cajani* inoculated seedlings of susceptible (ICP 7119) and moderately resistant (ICPL 20135, ICPL 99010, ICPL 99048) pigeonpea cultivars under controlled conditions [26]. Self-crossed breeder seeds of these cultivars were obtained from the pigeonpea breeding program at ICRISAT, Patancheru, India.

**Table 1.** Details of anti-oomycetes fungicides and their effect on *P. cajani*.

| Trade Name | Active Ingredient | Target Site | Manufacturer | FRAC Code [a] | Range of Dosage Tested (μg/mL) | MIC [b] | EC$_{50}$ [c] | Dose Recommended (kg/ha) |
|---|---|---|---|---|---|---|---|---|
| Acrobat® | Metiram 44% + Dimethomorph 9% WG | Multi-site contact activity; phospholipid biosynthesis and cell wall deposition | BASF India limited, Hyderabad, Telangana, India | M03 and 40 | 0.1–0.75 | 0.5 | 0.17 | 1.5 |
| Curzate® M8 | Cymoxanil 8% + Mancozeb 64% WP | Unknown; multi-site contact activity | E.I. DuPont India Pvt. Ltd., Mumbai-, Maharashtra, India | 27 and M03 | 0.1–100 | 60 | 8.23 | 1.5 |
| Equation® Pro | Famoxadone 16.6% + Cymoxanil 22.1% EC | Complex III of fungal respiration: ubiquinol oxidase; unknown | E.I. DuPont India Pvt. Ltd., Mumbai, Maharashtra, India | 11 and 27 | 0.1–140 | 140 | 24.96 | 0.5 |
| Indofil M-45® | Mancozeb 75% WP | Multi-site contact activity | Indofil chemicals company, Thane, Maharashtra, India | M03 | 0.1–100 | 100 | 16.86 | 2.0 |
| RidomilGold® | Metalaxyl-M 4% + Mancozeb 64% WP | RNA polymerase I; Multi-site contact activity | Syngenta India limited, Pune, Maharashtra, India | 4 and M03 | 0.1–100 | 35 | 2.49 | 2.5 |

[a] FRAC = Fungicide Resistance Action Committee. [b] Minimal inhibitory concentration (MIC) of fungicides which inhibits 100% of radial mycelial growth compared to untreated control. [c] Effective concentration (EC$_{50}$) of fungicides which inhibits 50% of radial mycelial growth.

### 2.3. Inhibition of P. cajani Mycelial Growth

A mycelial agar plug (5 mm in diameter) from a 7-day-old actively grown *P. cajani* culture was placed at the center of a petri plate (90 mm) containing 15 mL of tomato extract agar amended with different concentrations of test fungicides (Supplementary Table S1). Fungicides were added to the media after autoclaving once the media temperature reached ~55 °C. A total of five replicates for each concentration were maintained. The plates containing only *tomato extract agar,* without fungicides, were considered as controls. The plates were sealed with paraffin film and *incubated at* 25 ± 1 °C with 12 h light/dark conditions.

The colony diameter (mm) of *P. cajani* in each plate was measured in two perpendicular directions when the mycelial growth had reached the periphery in the control petri plate. Two diameters from perpendicular measurements were averaged after subtracting the diameter of the mycelial plug inoculum. The experiments were conducted twice under similar conditions. Data from two experiments were pooled together, and the effective concentration (EC$_{50}$) of each fungicide against *P. cajani* was determined [27].

### 2.4. Effect of Fungicides on Sporangia Formation and Zoospore Discharge

The efficacy of fungicides on the host-independent stages of *P. cajani* was determined using the following parameters: total number of sporangia developed, number of viable and non-viable sporangia, total number of zoospores discharged, and number of motile and encysted zoospores. The protocol developed by Sharma and Ghosh [20] for sporangia and zoospore production was followed to assess the above parameters. The mycelial plug

(5 mm) was added into a 100 mL conical flask containing 25 mL of 20% tomato extract broth, then incubated at $25 \pm 1\,^{\circ}$C under dark conditions for 72 h. After 72 h incubation, the tomato extract broth was decanted, and 25 mL of fungicides mixed with sterilized pond water was added during each water replacement step. The $EC_{50}$ values of each fungicide (Table 1) were selected to evaluate the effect on sporangia formation and zoospore discharge. The timing of fungicide application to the sporangia and zoospore production media is outlined in Figure 2, and the details are as follows:

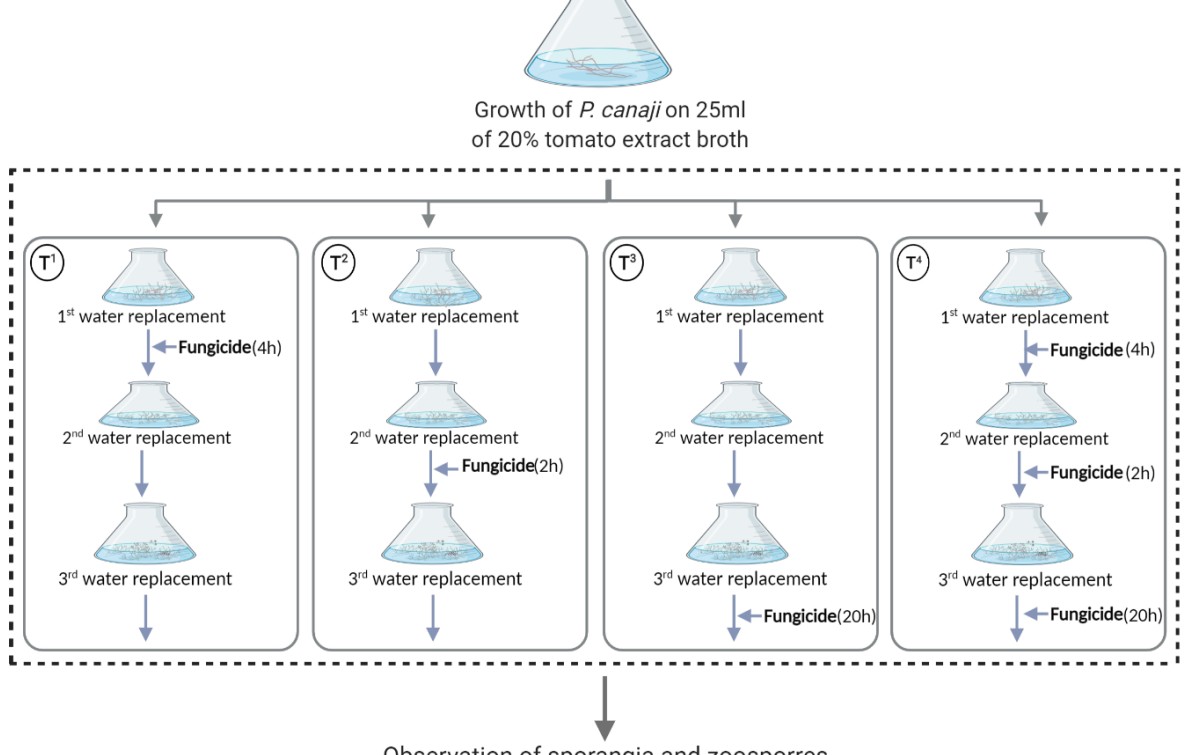

**Figure 2.** Schematic illustration of the time of fungicide applications to the media for *P. cajani* sporangia and zoospore induction. $T^1$—fungicide added during 1st water replacement (72nd–76th h); T2—fungicide added during 2nd water replacement (76th–78th h); T3—fungicide added during 3rd water replacement (78th–98th h); and T4—fungicide applied at all three steps during water replacement (72nd–98th h).

$T^1$ = fungicide added at step 1 for 4 h (72nd–76th h);
$T^2$ = fungicide added at step 2 for 2 h (76th–78th h);
$T^3$ = fungicide added at step 3 for 20 h (78th–98th h);
$T^4$ = fungicide added at all three steps for 26 h (72nd–98th h);
Control (without any fungicide).

Each treatment was replicated five times. The effects of fungicides on the above parameters were examined after 18 h of incubation under a compound microscope (Olympus CX41, Olympus Corporation, Shinjuku, Japan), then counted on a hemocytometer (Sigma-Aldrich, St. Louis, MO, USA) by averaging 8–10 microscopic fields in each replication.

### 2.5. Efficacy of Fungicides and Fungicide Application Methods in Disease Control

The efficacy of the fungicides and their different application methods were evaluated to control the Phytophthora blight of pigeonpea under glasshouse conditions [20]. Five fungicides were applied to the pigeonpea seeds and seedlings in different combinations: seed-treatment (ST), soil-drench (SD), and foliar-spray (FS). The detailed application methods were as follows:

Seed-treatment: before sowing;
Soil-drench: 24 h before zoospore inoculation;
Foliar-spray: 24 h after zoospore inoculation;
Seed-treatment + soil-drench;
Soil-drench + foliar-spray.

Two times the $EC_{50}$ value of each fungicide was used for seed-treatment and soil-drenching, and the $EC_{50}$ value of each fungicide was used for foliar-spray. Fungicide-treated susceptible ICP 7119 seeds (72 seeds per replication) were sown in plastic trays ($35 \times 25 \times 8$ cm$^3$) filled with a mixture of sterilized river sand and vermiculite (10:1 *v/v*) under natural light conditions ($28 \pm 2$ °C). Afterward, 7-day-old seedlings were inoculated with diluted zoospore ($1.5 \times 10^5$/mL) suspensions (~2 ml of zoospore suspension per plant). Inoculated seedling trays were kept in a glasshouse at $28 \pm 2$ °C under natural light conditions. Seedlings were flooded with sterilized water for 48 h, and the saturation level was maintained thereafter until the completion of the experiment. A similar number of seedling trays inoculated with only sterilized water and without fungicide served as negative controls, whereas the zoospore-inoculated seedling trays without fungicide treatments served as positive controls. Each treatment was replicated three times. The whole experiment was repeated thrice. The development of disease symptoms, i.e., appearance of small lesions on the stem, was monitored daily, and the total number of both infected plants and healthy plants was recorded. Disease severity, in terms of per cent disease incidence, was recorded at 7 days after inoculation by following the formula below [26].

$$\text{Per cent disease incidence} = \frac{\text{Number of infected plants}}{\text{Number of plants inoculated}} \times 100$$

Based on the above experimental results, effective fungicides (metiram + dimethomorph, cymoxanil + mancozeb, mancozeb, and metalaxyl-M + mancozeb) and the best fungicide application methods (soil-drench, seed-treatment + soil-drench; soil-drench + foliar-spray) were tested against susceptible (ICP 7119) and moderately resistant (ICPL 20135, ICPL 99010, ICPL 99048) cultivars. The experiments were repeated to assess the protective and curative properties of these fungicides in controlling Phytophthora blight in cultivars that are partially resistant to it.

### 2.6. Experimental Design and Statistical Data Analysis

The data from replicated experiments were combined for analysis. The $EC_{50}$ value of fungicides was estimated by fitting linear regression lines of probit-transformed inhibition data against the $log_{10}$-transformed fungicide concentration [28] in the R statistical program (R, Development Core Team, 2020). The data regarding sporangia formation and zoospore discharge were square-root transformed, and the data regarding the per cent disease incidence in fungicide-treated pigeonpea seedlings were first arcsine-transformed to normalize the residuals, and then back-transformed for the presentation of the results [29]. The transformed data were subjected to analysis of variance (ANOVA) to test for significant differences between the treatments and fungicide application methods in the R statistical program. The significance of mean differences within treatments and fungicide application methods was tested by Duncan's multiple range test at a $p < 0.01$ level of probability.

## 3. Result

### 3.1. Efficacy of Fungicides on Mycelial Growth of P. cajani

A wide range of fungicide concentrations were tested to determine the minimum inhibitory concentration (MIC) of fungicides needed to control the mycelial growth of *P. cajani*. The growth of *P. cajani* differed for different fungicides according to their active ingredients and different concentrations. Depending upon the inhibitory competence, the test concentration of fungicides ranged from 0.1 to 140 µg/mL (Supplementary Table S1). Among the five fungicides tested, metiram + dimethomorph was the most effective in controlling the *P. cajani* mycelia and showed the lowest MIC value (0.5 µg/mL), followed

by metalaxyl-M + mancozeb (35 μg/mL) and cymoxanil + mancozeb (60 μg/mL). On the other hand, famoxadone + cymoxanil showed a maximum MIC value of 140 μg/mL, followed by mancozeb (100 μg/mL) (Supplementary Table S1).

The mycelium growth of *P. cajani* was analyzed at different concentrations of each fungicide to determine the $EC_{50}$ values of the fungicides. metiram + dimethomorph showed the lowest $EC_{50}$ value (0.17 μg/mL) among all of the fungicides, followed by metalaxyl-M + mancozeb (2.49 μg/mL) and cymoxanil + mancozeb (8.23 μg/mL). On the contrary, famoxadone + cymoxanil (24.96 μg/mL) showed the highest $EC_{50}$ value, followed by mancozeb (16.86 μg/mL) (Table 1).

### 3.2. Efficacy of Fungicides on Sporangia Induction

The effect of fungicides on sporangia formation in *P. cajani* significantly differed according to the time of application ($p < 0.001$) (Supplementary Table S2). Irrespective of the time of fungicide application, metiram + dimethomorph and metalaxyl-M + mancozeb were found to be most effective in inhibiting sporangial development, followed by cymoxanil + mancozeb respectively (Table 2). On the other hand, famoxadone + cymoxanil and mancozeb showed the weakest effect in terms of inhibiting the sporangial development of *P. cajani*. Concisely, the metalaxyl-M + mancozeb and metiram + dimethomorph fungicides were found to be the most effective when they were added to the media for 4 h (from 72nd–76th h) at the $T^1$ step. Further, cymoxanil + mancozeb and mancozeb showed the maximum inhibition of sporangia development at the $T^3$ step (fungicide was in the media for 20 h; i.e., 78th–98th h) as compared to others. On the other hand, complete inhibition of sporangia was achieved by metiram + dimethomorph, metalaxyl-M + mancozeb and mancozeb at the $T^4$ step (fungicide was applied at all three steps). A reduction of approximately 88.31% was observed in sporangia formation in famoxadone + cymoxanil ($3.00 \pm 1.00$) compared to the control ($25.67 \pm 3.5$) (Table 2). Overall, metiram + dimethomorph, metalaxyl-M + mancozeb, and cymoxanil + mancozeb significantly decreased the number of sporangia, including viable/non-viable and abnormal sporangia formation, as compared to the mancozeb and famoxadone + cymoxanil fungicides. The application of fungicides at step $T^4$, as well as at $T^3$, $T^1$, and $T^2$, achieved the maximum reduction in sporangia development in the host-independent stages of *P. cajani*.

**Table 2.** Effect of fungicides and time of fungicide application on *P. cajani* sporangia development.

| Fungicides [a] | Treatment | Total Number of Sporangia [b] | Reduction in Sporangia (%) [c] | Viable Sporangia | Non-Viable Sporangia | Abnormal Sporangia |
|---|---|---|---|---|---|---|
| Metiram 44% + Dimethomorph 9% WG | $T^1$ | $1.67 \pm 0.58$ [d] | 93.51 | $0.67 \pm 0.58$ | $0.67 \pm 0.58$ | $0.33 \pm 0.58$ |
| | $T^2$ | $2.00 \pm 1.00$ | 92.21 | $1.00 \pm 0.00$ | $0.67 \pm 0.58$ | $0.33 \pm 0.58$ |
| | $T^3$ | $2.03 \pm 0.62$ | 92.09 | $0.67 \pm 0.58$ | $0.67 \pm 0.58$ | $0.00 \pm 0.00$ |
| | $T^4$ | $0.00 \pm 0.00$ | 100.00 | $0.00 \pm 0.00$ | $0.00 \pm 0.00$ | $0.00 \pm 0.00$ |
| Cymoxanil 8% + Mancozeb 64% WP | $T^1$ | $4.00 \pm 1.00$ | 84.42 | $2.67 \pm 1.53$ | $1.00 \pm 1.00$ | $0.33 \pm 0.58$ |
| | $T^2$ | $4.67 \pm 1.15$ | 81.82 | $3.00 \pm 1.00$ | $1.00 \pm 1.00$ | $0.67 \pm 0.58$ |
| | $T^3$ | $0.00 \pm 0.00$ | 100.00 | $0.00 \pm 0.00$ | $0.00 \pm 0.00$ | $0.00 \pm 0.00$ |
| | $T^4$ | $0.00 \pm 0.00$ | 100.00 | $0.00 \pm 0.00$ | $0.00 \pm 0.00$ | $0.00 \pm 0.00$ |
| Famoxadone 16.6% + Cymoxanil 22.1% EC | $T^1$ | $18.33 \pm 7.64$ | 28.58 | $13.33 \pm 5.51$ | $3.33 \pm 1.15$ | $1.67 \pm 0.58$ |
| | $T^2$ | $8.67 \pm 1.53$ | 66.24 | $5.00 \pm 1.00$ | $2.33 \pm 1.53$ | $1.33 \pm 0.58$ |
| | $T^3$ | $7.67 \pm 1.53$ | 70.13 | $5.67 \pm 1.53$ | $1.67 \pm 0.58$ | $0.33 \pm 0.58$ |
| | $T^4$ | $3.00 \pm 1.00$ | 88.31 | $1.67 \pm 1.15$ | $1.00 \pm 1.00$ | $0.33 \pm 0.58$ |

| Fungicides [a] | Treatment | Total Number of Sporangia [b] | Reduction in Sporangia (%) [c] | Viable Sporangia | Non-Viable Sporangia | Abnormal Sporangia |
|---|---|---|---|---|---|---|
| Mancozeb 75% WP | $T^1$ | $18.33 \pm 2.52$ | 28.58 | $9.33 \pm 1.53$ | $6.33 \pm 2.08$ | $2.67 \pm 1.53$ |
| | $T^2$ | $12.67 \pm 1.53$ | 50.66 | $7.00 \pm 4.58$ | $3.33 \pm 1.53$ | $2.33 \pm 0.58$ |
| | $T^3$ | $0.33 \pm 0.58$ | 98.70 | $0.33 \pm 0.58$ | $0.00 \pm 0.00$ | $0.00 \pm 0.00$ |
| | $T^4$ | $0.00 \pm 0.00$ | 100.00 | $0.00 \pm 0.00$ | $0.00 \pm 0.00$ | $0.00 \pm 0.00$ |
| Metalaxyl-M 4% + Mancozeb 64% WP | $T^1$ | $1.33 \pm 0.58$ | 94.81 | $0.67 \pm 0.58$ | $0.33 \pm 0.58$ | $0.33 \pm 0.58$ |
| | $T^2$ | $2.67 \pm 0.58$ | 89.61 | $1.67 \pm 0.58$ | $0.67 \pm 0.58$ | $0.33 \pm 0.58$ |
| | $T^3$ | $2.33 \pm 0.58$ | 90.91 | $1.00 \pm 0.00$ | $1.00 \pm 1.00$ | $0.67 \pm 0.58$ |
| | $T^4$ | $0.00 \pm 0.00$ | 100.00 | $0.00 \pm 0.00$ | $0.00 \pm 0.00$ | $0.00 \pm 0.00$ |
| Control | - | $25.67 \pm 3.51$ | - | $0.00 \pm 0.00$ | $0.00 \pm 0.00$ | $0.00 \pm 0.00$ |

[a] An amount of fungicide equal to the $EC_{50}$ value (for details, see Table 1) was added to the media used for *P. cajani* growth. $T^1$, fungicide applied at the time of step $T^1$ for 4 h (72nd–76th h); $T^2$, fungicide applied at the time of step $T^2$ for 2 h (76th–78th h); $T^3$, fungicide applied at the time of step $T^3$ for 20 h (78th–98th h); and $T^4$, fungicide applied at all three steps for 26 h (72nd–98th h) (for details, refer to Figures 1 and 2). [b] Zoospores present in 10 μL of suspension. [c] Reduction in zoospores (%) was calculated compared to the control (media without fungicide). [d] Mean $\pm$ standard deviations; '-': not applicable. For the statistical significance of the individual parameters of the zoospores, see Supplementary Table S2.

*3.3. Efficacy of Fungicides on Zoospore Induction*

The zoospores' development (number of zoospores, motility, encystment, etc.) and germination were considerably affected by fungicides and their time of application ($p < 0.001$) (Supplementary Table S2). The metiram + dimethomorph fungicide ($1.67 \pm 1.53$) most significantly inhibited the formation of zoospores, followed by the metalaxyl-M + mancozeb ($2.00 \pm 1.00$) and cymoxanil + mancozeb ($8.00 \pm 1.00$), at the $T^1$ step (Table 3). Although, metiram + dimethomorph was effective at the $T^1$ step in reducing zoospore numbers, it was not effective in reducing zoospore motility or inhibiting zoospore germination. The application of fungicide at step $T^3$ for 20 h (78th–98th h) resulted in a 100% reduction in zoospore formation by the metiram + dimethomorph, metalaxyl-M + mancozeb, and mancozeb fungicides. In contrast, at the $T^4$ step (72nd–98th h), the lowest level of zoospore inhibition was recorded in famoxadone + cymoxanil (98.11%), even though the remaining fungicides showed 100% reductions in zoospore development and germination compared to the untreated control ($35.33 \pm 3.06$). The application of fungicides to the zoospore induction media at the $T^4$ step followed by the $T^3$ step significantly inhibited the formation of zoospores, as well as their motility, encystment, and germination. However, the least significant effect of zoospore formation was noticed with famoxadone + cymoxanil, and the rest of the fungicides showed a significant reduction in zoospores as compared to the untreated controls.

**Table 3.** Effect of fungicides and time of fungicide application on *P. cajani* zoospore development.

| Fungicides [a] | Treatment | Total Number of Zoospore [b] | Reduction in Zoospore (%) [c] | Motile Zoospore | Encysted Zoospore | Germinated Zoospore |
|---|---|---|---|---|---|---|
| Metiram 44% + Dimethomorph 9% WG | $T^1$ | $1.67 \pm 1.53$ [d] | 95.28 | $0.67 \pm 1.15$ | $0.67 \pm 0.58$ | $0.33 \pm 0.58$ |
| | $T^2$ | $2.33 \pm 0.58$ | 93.40 | $1.00 \pm 1.00$ | $1.33 \pm 0.58$ | $0.00 \pm 0.00$ |
| | $T^3$ | $0.00 \pm 0.00$ | 100.00 | $0.00 \pm 0.00$ | $0.00 \pm 0.00$ | $0.00 \pm 0.00$ |
| | $T^4$ | $0.00 \pm 0.00$ | 100.00 | $0.00 \pm 0.00$ | $0.00 \pm 0.00$ | $0.00 \pm 0.00$ |

**Table 3.** *Cont.*

| Fungicides [a] | Treatment | Total Number of Zoospore [b] | Reduction in Zoospore (%) [c] | Motile Zoospore | Encysted Zoospore | Germinated Zoospore |
|---|---|---|---|---|---|---|
| Cymoxanil 8% + Mancozeb 64% WP | T[1] | $8.00 \pm 1.00$ | 77.36 | $2.67 \pm 0.58$ | $5.33 \pm 0.58$ | $0.00 \pm 0.00$ |
| | T[2] | $9.67 \pm 1.15$ | 72.64 | $1.00 \pm 0.00$ | $8.33 \pm 1.53$ | $0.33 \pm 0.58$ |
| | T[3] | $0.00 \pm 0.00$ | 100.00 | $0.00 \pm 0.00$ | $0.00 \pm 0.00$ | $0.00 \pm 0.00$ |
| | T[4] | $0.00 \pm 0.00$ | 100.00 | $0.00 \pm 0.00$ | $0.00 \pm 0.00$ | $0.00 \pm 0.00$ |
| Famoxadone 16.6% + Cymoxanil 22.1% EC | T[1] | $17.67 \pm 1.15$ | 50.00 | $15.00 \pm 2.65$ | $2.67 \pm 1.53$ | $0.00 \pm 0.00$ |
| | T[2] | $20.67 \pm 2.52$ | 41.50 | $16.33 \pm 2.52$ | $4.33 \pm 0.58$ | $0.00 \pm 0.00$ |
| | T[3] | $3.00 \pm 1.00$ | 91.51 | $0.00 \pm 0.00$ | $3.00 \pm 1.00$ | $0.00 \pm 0.00$ |
| | T[4] | $0.67 \pm 1.15$ | 98.11 | $0.33 \pm 0.58$ | $0.33 \pm 0.58$ | $0.00 \pm 0.00$ |
| Mancozeb 75% WP | T[1] | $5.00 \pm 2.00$ | 85.85 | $1.00 \pm 0.00$ | $4.00 \pm 2.00$ | $0.00 \pm 0.00$ |
| | T[2] | $11.67 \pm 2.52$ | 66.98 | $6.33 \pm 1.53$ | $4.33 \pm 1.53$ | $1.00 \pm 1.00$ |
| | T[3] | $4.33 \pm 1.15$ | 87.73 | $1.33 \pm 0.58$ | $3.00 \pm 1.00$ | $0.00 \pm 0.00$ |
| | T[4] | $0.00 \pm 0.00$ | 100.00 | $0.00 \pm 0.00$ | $0.00 \pm 0.00$ | $0.00 \pm 0.00$ |
| Metalaxyl-M 4% + Mancozeb 64% WP | T[1] | $2.00 \pm 1.00$ | 94.34 | $1.33 \pm 0.58$ | $0.67 \pm 0.58$ | $0.00 \pm 0.00$ |
| | T[2] | $2.00 \pm 2.00$ | 94.34 | $0.00 \pm 0.00$ | $2.00 \pm 2.00$ | $0.00 \pm 0.00$ |
| | T[3] | $0.00 \pm 0.00$ | 100.00 | $0.00 \pm 0.00$ | $0.00 \pm 0.00$ | $0.00 \pm 0.00$ |
| | T[4] | $0.00 \pm 0.00$ | 100.00 | $0.00 \pm 0.00$ | $0.00 \pm 0.00$ | $0.00 \pm 0.00$ |
| Control | - | $35.33 \pm 3.06$ | - | $35.33 \pm 3.06$ | - | - |

[a] An amount of fungicide equal to the $EC_{50}$ value (for details, see Table 1) was added to the media used for *P. cajani* growth. T[1], fungicide applied at the time of step T[1] for 4 h (72nd–76th h); T[2], fungicide applied at the time of step T[2] for 2 h (76th–78th h); T[3], fungicide applied at the time of step T[3] for 20 h (78th–98th h); and T[4], fungicide applied at all three steps for 26 h (72nd–98th h) (for details, refer to Figures 1 and 2). [b] Zoospores present in 10 μL of suspension. [c] Reduction in zoospores (%) was calculated compared to the control (media without fungicide). [d] Mean $\pm$ standard deviations; '-': not applicable. For the statistical significance of individual parameters of zoospores, see Supplementary Table S2.

*3.4. Efficacy of Fungicide and Fungicide Application Methods on Suppression of Phytophthora Blight at the Seedling Stage*

Zoospore infections and associated plant death was first observed in the inoculated control, followed by the fungicide treatments (Figure 3). The fungicide application methods and the fungicides themselves were significant at $p < 0.001$ and $p < 0.05$, whereas interactions (methods × fungicides) were not significant (Table 4). Among the fungicide application methods, the lowest per cent disease incidence was noticed with seed-treatment + soil-drench treatment, followed by soil-drench + foliar-spray and soil-drench alone. The maximum per cent disease incidence was observed with foliar-spray of fungicides, followed by seed-treatment (Figure 4). Among the fungicides, the lowest disease incidence was noticed with seed-treatment + soil-drench using metalaxyl-M + mancozeb (6.6%) and with soil-drench + foliar-spray using metalaxyl-M + mancozeb (8.6%). The metiram + dimethomorph (11.6), mancozeb (11.3%) and cymoxanil + mancozeb (15.8%) fungicides were on par with each other ($p < 0.05$) concerning inhibition of the Phytophthora blight using the seed-treatment + soil-drench method. (Supplementary Table S3). However, soil-drenching with metalaxyl-M + mancozeb (21.9%) showed the lowest per cent disease incidence as compared to the maximum, found with the famoxadone + cymoxanil (51.30%) fungicide. The seed-treatment + foliar-spray method, using different fungicides, were on par with each other at $p < 0.05$, respectively.

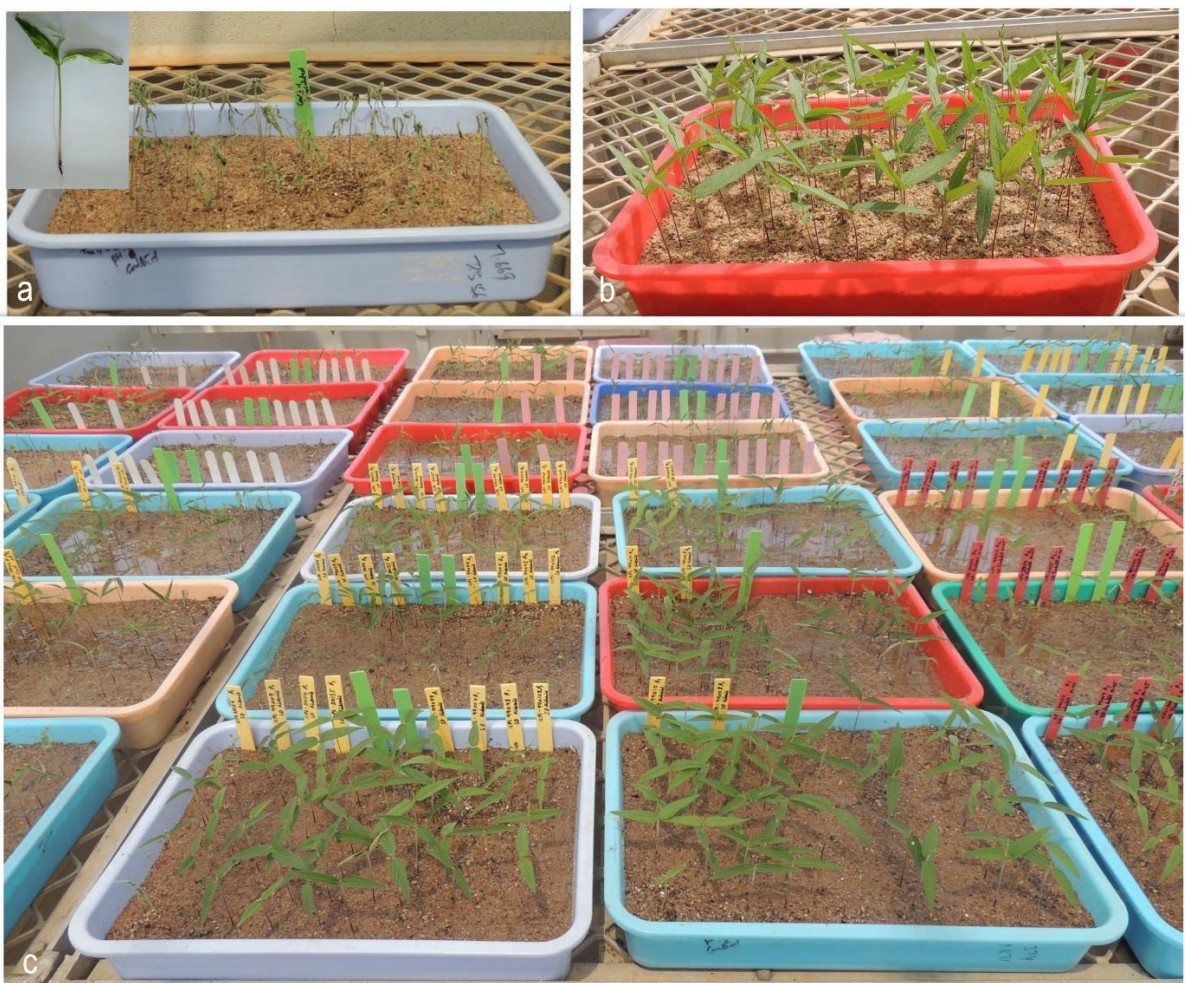

**Figure 3.** Evaluation of fungicide efficacy against Phytophthora blight of pigeonpea seedlings. (**a**) Zoospore-inoculated positive control; (**b**) non-inoculated negative control. (**c**) Overview of the zoospore-inoculated seedlings with different fungicide treatments.

**Table 4.** Analysis of variance (ANOVA) evaluating the effects of different fungicides and their different application methods for controlling the Phytophthora blight of pigeonpea seedlings in a susceptible ICP 7119 cultivar.

| Source of Variation | Df | Sum Sq | Mean Sq | F Value | Pr (>F) |
|---|---|---|---|---|---|
| Methods [⊥] | 4 | 12,434.2 | 3108.56 | 17.9364 | $4.62 \times 10^{-7}$ *** |
| Fungicide [#] | 4 | 2259.4 | 564.86 | 3.2593 | 0.02784 * |
| Methods × Fungicide | 16 | 1638.5 | 102.4 | 0.5909 | 0.8615 [ns] |
| Residuals | 25 | 4332.8 | 173.31 | | |

[⊥] Five different fungicide application methods (seed-treatment; soil-drench; foliar-spray; seed-treatment + soil-drench; and soil-drench + foliar-spray). [#] Five fungicides (metiram + dimethomorph, cymoxanil + mancozeb, famoxadone + cymoxanil, mancozeb, and metalaxyl-M + mancozeb). Df: degrees of freedom. Significant codes: *** 0.001; '*' 0.05; [ns] non-significant.

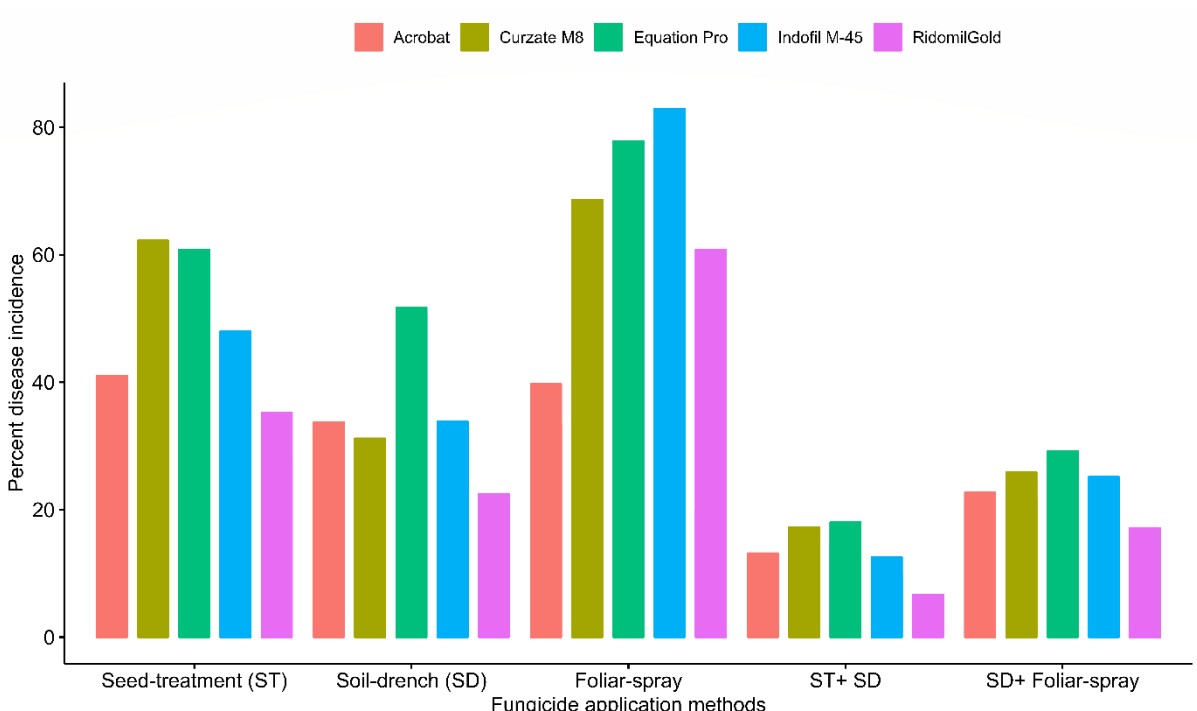

**Figure 4.** Evaluation of fungicide efficacy and application methods in controlling Phytophthora blight of pigeonpea seedlings on susceptible ICP 7119 cultivar.

Based on the above experimental results, different fungicide application methods, seed-treatment + soil-drench, soil-drench + foliar-spray, and soil-drench alone were selected to test the efficacy of the metiram + dimethomorph, metalaxyl-M + mancozeb, mancozeb, and metalaxyl-M + mancozeb fungicides in pigeonpea ICPL 99010, ICPL 20135, ICPL 99048, and ICP 7119 cultivars. The analysis of variance (ANOVA) showed that the fungicide application methods, fungicides, and cultivars, as well as the interactions of the methods with the fungicides and the methods with the cultivars, were significant at $p < 0.001$. However, the interactions of the fungicides with the cultivars were not significant, although the interaction of the methods with the fungicides and cultivars were significant, at $p < 0.05$ (Supplementary Table S4. Among the fungicide application methods, seed-treatment + soil-drench achieved the lowest per cent disease incidence in different pigeonpea cultivars as compared to the soil-drench method alone (Figure 5a). Between the fungicides, least per cent disease incidence was noticed with metalaxyl-M + mancozeb, followed by metiram + dimethomorph, mancozeb, and cymoxanil + mancozeb (Figure 5b). When we compare the fungicides and different fungicide application methods with the various cultivars, the minimum per cent disease incidence was observed in ICPL 99010, followed by the ICPL 88048 and ICPL 20135 cultivars. The maximum per cent disease incidence was recorded in the susceptible ICP 7119 cultivar (Supplementary Table S5).

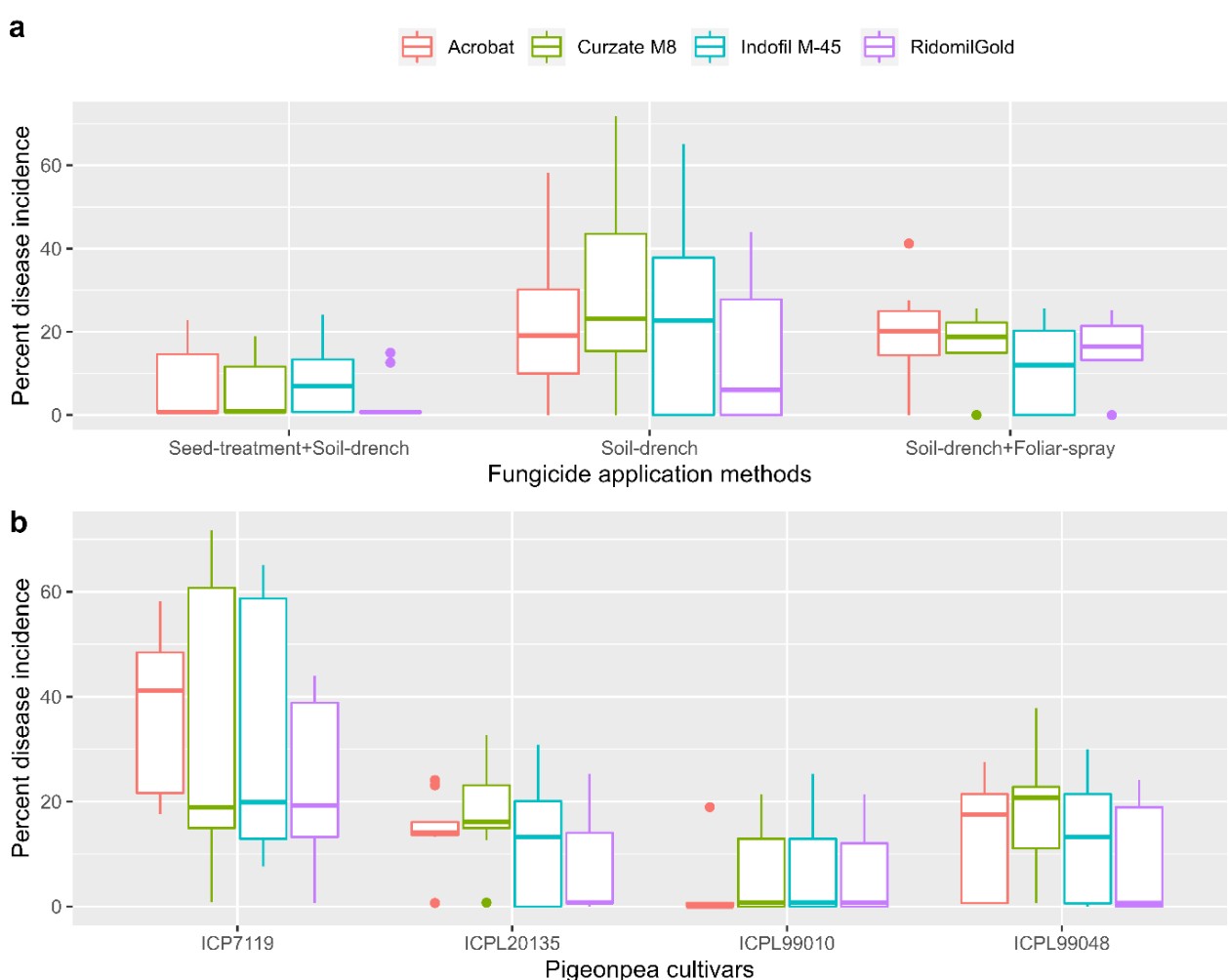

**Figure 5.** Evaluation of different fungicides and fungicide application methods for controlling the Phytophthora blight in different pigeonpea cultivars. (**a**) Fungicide application methods; (**b**) different pigeonpea cultivars.

## 4. Discussion

*P. cajani* is largely dissimilar to other oomycetes concerning the optimum conditions for its growth and development [8]. The incidence of Phytophthora blight is more severe in the seedling stage as compared to other crop growth stages [18]. Good agricultural practices, such as planting healthy seeds and using effective field sanitization practices, may reduce the incidence of the disease [20]. Moreover, under favorable weather conditions, cultural practices and the use of available, partially disease-resistant sources may lead to failure to control the *P. cajani* infection and its spread [1,10,29]. Moreover, no report exists on the true host resistance of pigeonpea to Phytophthora blight. Hence, there is a dependency on fungicides for managing this disease in farming fields. Most of the previous studies on fungicidal effects on *P. cajani* have focused solely on mycelium inhibition, pot, and/or natural field screening techniques with old molecules [30]. Thus, it has now become very important to evaluate the effects of new fungicidal molecules on *P. cajani*. We believe our study to be the first evaluation of a major class of fungicides and the methods of their application against sporangia and zoospore induction, as well as the management of a seedling blight, under glasshouse conditions.

We selected the fungicides based on their modes of action and targeted impacts on the fungus. Of the five fungicides tested, the MIC and $EC_{50}$ values for the inhibition of radial mycelia were the maximum in the metiram + dimethomorph, metalaxyl-M + mancozeb, and cymoxanil + mancozeb fungicides. The multi-site contact activity, phospholipid

biosynthesis, and cell wall deposition of metiram + dimethomorph; the multi-site contact activity, nucleic acid synthesis, and RNA polymerase of metalaxyl-M + mancozeb; and the multi-site contact activity and control of post-infection fungal activity of cymoxanil + mancozeb all significantly reduced the mycelial growth of *P. cajani*. Combinations of metiram and dimethomorph, mancozeb and metalaxyl-M, mancozeb and cymoxanil, and cymoxanil and famoxadone are very effective in the inhibition of *Phytophthora* spp. mycelia in various crops [31–33].

Both sporangium formation and zoospore germination are important in initiating infection and spread of plant pathogens [34]. Elliott et al. [32] observed that in the case of *P. ramorum*, systemic fungicides were more effective than contact fungicides in preventing sporangia and zoospore germination. In our study, we observed that the metalaxyl-M + mancozeb, metiram + dimethomorph, and cymoxanil + mancozeb fungicides have intrinsic inhibitory activity on sporangia viability, zoospore germination, and encystment of *P. cajani*, whereas famoxadone + cymoxanil and mancozeb have multi-site contact activity, but less of an impact on *P. cajani*. Cohen and Gisi [35] noticed that temporary exposure to fungicide for up to an hour was not detrimental to fungal spore germination nor the infectivity of sporangia or cystospores, but it inhibited their further growth and deformed their shape. The fungicide application stage and total incubation period are crucial factors in inhibiting sporangia and zoospore formation. The constant availability of fungicides during sporangia and zoospore induction at step $T^3$ for 20 h (78th–98th h) had a greater impact on the inhibition of sporangia and zoospores. The application of fungicides at the time of sporangiophore formation is more significant in controlling the zoospores than applying the fungicides at the mycelial growth stage.

Combinations of the methods of seed-treatment + soil-drench, soil-drench + foliar-spray, and soil-drench alone, were effective in controlling the Phytophthora blight at the seedling stage on a susceptible ICP 7119 cultivar. Overall, the metalaxyl-M + mancozeb, metiram + dimethomorph, mancozeb, and cymoxanil + mancozeb fungicides were very effective in reducing the progression and symptoms associated with the *P. cajani* disease in different pigeonpea cultivars. These fungicides may be able to stop zoospore infection by controlling the post-infection activity of the fungus at the seedling stage. Seed treatment with metalaxyl [11], including a combination of seed-dressing and foliar-spray with metalaxyl, was effective in controlling the Phytophthora pigeonpea blight [36]. An integration of *P. fluorescens* with an apron or RidomilGold MZ as seed treatment significantly reduced the incidence of the Phytophthora blight in pigeonpea plants, and enhanced seed germination and grain yield [23]. Kannaiyan and Nene [11] reported that foliar-spray with fungicide had the lowest impact on the Phytophthora blight in pigeonpea plants. In our in vivo study, we also observed similar results; the foliar-spray of fungicides had the least significant effect in controlling seedling blights in pigeonpea plants. Previous glasshouse experiments indicate that, irrespective of crops, 90% of Phytophthora blight occurrences can be controlled by metalaxyl [30]. Our results are in agreement with previous reports affirming that seed-treatment and soil-drenching with fungicides are more effective methods foliar spray in controlling the Phytophthora disease in many crops [31,34,37].

In conclusion, novel broad-spectrum anti-oomycetes combi-fungicides are most effective in inhibiting mycelial growth and sporangia and zoospore germination in *P. cajani*. Our study provides new insights into the control of sporangia formation and zoospore discharge. From a practical point of view, the combination of seed-treatment + soil-drench using the metalaxyl-M + mancozeb and metiram + dimethomorph fungicides on partially resistant (ICPL 99010, ICPL 88048 and ICPL 20135) cultivars could provide additive or synergistic effects in the control of Phytophthora blight at the seedling stage. In addition, there is a need to assess the effects of novel fungicides under natural and artificial field conditions to prepare to manage the Phytophthora blight in pigeonpea at the flowering and maturity stages of the crop.

**Supplementary Materials:** The following supporting information can be downloaded at: https:// www.mdpi.com/article/10.3390/agriculture13030633/s1, Table S1: Evaluation of different fungicides concentrations on radial growth of *P. cajani*; Table S2:Summary of analysis of variance (ANOVA) of different fungicides (A) and their time of applications (B) during sporangia and zoospore formation in *P. cajani* under in vitro condition; Table S3: Effect of different fungicides and fungicide application methods on controlling of Phytophthora blight on susceptible ICP 7119 cultivar; Table S4: Analysis of variance (ANOVA) for evaluation of different fungicides and their different application methods for controlling of Phytophthora blight in different pigeonpea cultivars; Table S5: Evaluation of Fungicide application methods and fungicides to manage the Phytophthora blight on different pigeonpea cultivars.

**Author Contributions:** M.S. conceived the project and designed the experiments; R.G. and A.T. performed the experiments; R.G. analyzed the data; R.G., M.S. and A.T. wrote the paper. All authors have read and agreed to the published version of the manuscript.

**Funding:** This research was partially supported by National Food Security Mission (NFSM) government of India under a project entitled "Addressing Phytophthora blight of pigeonpea: as emerging disease treat to pigeonpea"; and Department of Science and Technology, Climate Change Program, government of India, under the project entitled "Center of Excellence on Climate Change Research for Plant Protection (CoE-CCRPP): Pest and disease management for climate change adaptation", grant number [DST/CCP/CoE/142/2018(G)].

**Institutional Review Board Statement:** Not applicable.

**Informed Consent Statement:** Not applicable.

**Data Availability Statement:** The data that support the findings of this study are available from the corresponding author upon reasonable request.

**Acknowledgments:** The authors are thankful to Bal Krishna and K. Ramulu, Legume Pathology, ICRISAT, Patancheru, India for technical assistance.

**Conflicts of Interest:** The authors declare no conflict of interest.

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
