# Peer review of "Evaluation of Fungicides and Fungicide Application Methods to Manage Phytophthora Blight of Pigeonpea"

_agriculture, doi:10.3390/agriculture13030633_

Round 1
Reviewer 1 Report (New Reviewer)
Dear Authors, I appreciate your efforts in conducting these interesting experiments and preparing the manuscript. I would recommend it for publication, however, there are some serious flaws (please see my comments below). The manuscript is interesting and also fits the journal.
1. Please reduce the abstract's length. I recommend wisely selecting the sentences for the abstract. This is the summary of the manuscript.
2. Line 45: Please put a full stop symbol after "sterility mosaic disease [1]".
3. Lines 47–48: Incorrect sentence. Please change the existing sentence to "The manner of causation and condition of the disease were initially ascribed to the pathogen as Phytophthora drechsleri sp. cajani, but later it was described as Phytophthora cajani [5]."
4. Please change the introduction section. Currently, it looks too much like a general introduction. State of the art is lacking. A very small part describes the "Evaluation of Fungicides and Fungicide Application Methods."
5. Line 92: Please briefly mention the part of the plant, from which the sample was collected.
6. Line 93: Please mention the GPS location, from where the samples were collected.
7. Line 99: Please avoid referencing in the figure legend.
8. Line 104: Please mention the Company name and location of the fungicides.
9. Please make the line numbers continuous, this makes it very difficult for the reviewers to mention the exact line number.
10. For figures, the description must be below the figure.
11. Table 2: Have the authors conducted any statistical analyses? The authors have mentioned the variation of the sporangia formation. But what about the significance factors?
12. Table 2: in some cases (eg: Indofil M-45; T3; Total sporangia and a few others), the standard error is higher than the average value. What is the reason for this discrepancy?
13. Table 2 and 3: there are many cases where the Standard error is almost near the average value. I recommend increasing the number of replications.
14. Please show the significance values in figure 4
15. Page 15: Line 270: See-treatment or seed treatment?
16. A general suggestion: please put all the methodological figures in the supplementary, instead the authors can represent the results in graphical format. This makes the results interesting for the readers.
Author Response
Reviewer 1
Q1: Please reduce the abstract's length. I recommend wisely selecting the sentences for the abstract. This is the summary of the manuscript.
Ans: Thank you for the advice, reduced the abstract significantly around 250 words.
Q2: Line 45: Please put a full stop symbol after "sterility mosaic disease [1]".
Ans: Added
Q3: Lines 47–48: Incorrect sentence. Please change the existing sentence to "The manner of causation and condition of the disease were initially ascribed to the pathogen as Phytophthora drechsleri sp. cajani, but later it was described as Phytophthora cajani [5]."
Ans: Changed the sentence
Q4: Please change the introduction section. Currently, it looks too much like a general introduction. State of the art is lacking. A very small part describes the "Evaluation of Fungicides and Fungicide Application Methods."
Ans: Thank you for the comment, changes are made in the introduction.
Q5: Line 92: Please briefly mention the part of the plant, from which the sample was collected.
Ans: added the brief description of plant part
Q6: Line 93: Please mention the GPS location, from where the samples were collected.
Ans: as suggested GPS location is mentioned
Q7: Line 99: Please avoid referencing in the figure legend.
Ans: deleted the reference in the figure legend
Q8: Line 104: Please mention the Company name and location of the fungicides.
Ans; Its mentioned in the table 1
Q9: Please make the line numbers continuous, this makes it very difficult for the reviewers to mention the exact line number.
Ans: We do understand the situation, and we will inform to editor
Q10: For figures, the description must be below the figure.
Ans: I think this is MDPI standards
Q11: Table 2: Have the authors conducted any statistical analyses? The authors have mentioned the variation of the sporangia formation. But what about the significance factors?
Ans: Yes, we conducted statistical analysis, more results are mentioned in the attached supplementary table 2
Q12: Table 2: in some cases (eg: Indofil M-45; T3; Total sporangia and a few others), the standard error is higher than the average value. What is the reason for this discrepancy?
Ans:
Q13: Table 2 and 3: there are many cases where the Standard error is almost near the average value. I recommend increasing the number of replications.
Ans:
Q14: Please show the significance values in figure 4
Ans: mentioned in the results.
Q15: Page 15: Line 270: See-treatment or seed treatment?
Ans: it’s a seed-treatment
- A general suggestion: please put all the methodological figures in the supplementary, instead the authors can represent the results in graphical format. This makes the results interesting for the readers.
Ans: Thank you for the suggestion, since there is little work has been done on this pathogen, and may people in India are facing problems in isolation zoospore production. Therefore, this methodology is better to be publish upfront.

Reviewer 2 Report (New Reviewer)
The manuscript present that novel broad-spectrum anti-oomycetes combi-fungicides are effective in inhibition of Phytophthora cajani mycelial growth, sporangia and zoospore germination. The combination of seed-treatment and soil-drench RidomilGold and Acrobat fungicides on partially resistant (ICPL 99010, ICPL 88048 and ICPL 20135) cultivars can produce additive or synergistic effects in the control of Phytophthora blight at the seedling stage, which has practical value. In general, this is an interesting topic. However, there are many substantial errors need to be revised.
Please try to use new references. Only 11 references from the last decade are cited.
Author Response
Reviewer 2
The manuscript present that novel broad-spectrum anti-oomycetes combi-fungicides are effective in inhibition of Phytophthora cajani mycelial growth, sporangia and zoospore germination. The combination of seed-treatment and soil-drench RidomilGold and Acrobat fungicides on partially resistant (ICPL 99010, ICPL 88048 and ICPL 20135) cultivars can produce additive or synergistic effects in the control of Phytophthora blight at the seedling stage, which has practical value. In general, this is an interesting topic. However, there are many substantial errors need to be revised.
Q: Please try to use new references. Only 11 references from the last decade are cited.
Ans: Thank you for the suggestion, since there is very little information available on this pathogen we could add new reference, that is the reason we have old references. As need of the hour, this work direct the other researcher to work on this pathogen or disease.

Reviewer 3 Report (New Reviewer)
I have thoroughly reviewed the manuscript entitled "Evaluation of Fungicides and Fungicide Application Methods to Manage Phytophthora Blight of Pigeonpea" which described the management strategies for Phytophthora blight of pigeonpea caused by Phytophthora cajani. They concluded that Combination of seed-treatment+soil-drench and soil-drench+foliar spray of Mancozeb (64%) with Metalaxyl-M (4%), or Metiram (44%) with Dimethomorph (9%) on moderately resistant cultivars gives synergistic effect in controlling of Phytophthora blight at seed- 32 ling stage. The manuscript is interesting and written well. but there are some serious deficiencies which should be addressed before making the final decision.
1- English language should be improved. At many stages, there are long sentences, which made it difficult to understand. Authors must revise English carefully
2-. The introduction presents important information but lacks recent literature which has made it unfit as a paper with the current state of knowledge. Authors are suggested to give historic developments in the development of resistance and management of disease along with decent development and describe the gaps and add some novelty statements
3- Captions of tables and figures must be comprehensive and self-explanatory
4- The discussion section must be improved by adding the justification of results and authors must try to avoid repetition of results in this section
5- Add a separate heading for the discussion section.
Author Response
Reviewer 3
I have thoroughly reviewed the manuscript entitled "Evaluation of Fungicides and Fungicide Application Methods to Manage Phytophthora Blight of Pigeonpea" which described the management strategies for Phytophthora blight of pigeonpea caused by Phytophthora cajani. They concluded that Combination of seed-treatment+soil-drench and soil-drench+foliar spray of Mancozeb (64%) with Metalaxyl-M (4%), or Metiram (44%) with Dimethomorph (9%) on moderately resistant cultivars gives synergistic effect in controlling of Phytophthora blight at seed- 32 ling stage. The manuscript is interesting and written well. but there are some serious deficiencies which should be addressed before making the final decision.
Q1- English language should be improved. At many stages, there are long sentences, which made it difficult to understand. Authors must revise English carefully
Ans: thank you for the suggestion, revised the English wherever is necessary
Q2- The introduction presents important information but lacks recent literature which has made it unfit as a paper with the current state of knowledge. Authors are suggested to give historic developments in the development of resistance and management of disease along with decent development and describe the gaps and add some novelty statements
Ans; Thank you for the suggestion, since there is very little information available on this pathogen we could add new reference, that is the reason we have old references. As suggested changes are made in the manuscript
Q3- Captions of tables and figures must be comprehensive and self-explanatory
Ans: thank you for suggestion, we changed the legends, made self-explanatory
Q4- The discussion section must be improved by adding the justification of results and authors must try to avoid repetition of results in this section
Ans: thank you for comment, changed the text wherever its necessary
Q5- Add a separate heading for the discussion section.
Ans: thank you, the discussion heading already mentioned

Round 2
Reviewer 1 Report (New Reviewer)
The authors have improved the manuscript; however, I cannot see any answers to Q12 & Q13 in the "Author response to report 1" file. Additionally, the authors must check the manuscript thoroughly for language errors (for eg: on page 10, line 56: Two times of EC50 values of each fungicide atwas used for seed-treatment”; please correct “atwas”). Please also check for linguistic errors.
Author Response
Answers to Q12 & Q13
Thank you for the comment. we presented standard deviation in the manuscript, not standard error. We recorded zero viable and non-viable sporangia in few replications, therefore standard deviation is higher than the average value in few cases.
Thank you for the suggestion language errors, and we will opt for MDPI English language editing.
The authors thank the reviewer for their valuable inputs which has enhanced the quality of our manuscript.
This manuscript is a resubmission of an earlier submission. The following is a list of the peer review reports and author responses from that submission.
Round 1
Reviewer 1 Report
- Interoduction is poor and not updated.
- no molecular identification of the casual organism
- No robust NOVELTY.
- Old literature reviews.
Author Response
The authors thank the reviewers for their valuable inputs which has enhanced the quality of our manuscript. The suggested changes have been duly made to the best of our knowledge.
As suggested changed the Introduction and added the molecular identify of the pathogen.
Deleted the old reviews.
Since the pathogen is more confined to India, further research is needed to understand the consequence on yield loses.
Reviewer 2 Report
In this article, the authors evaluated five fungicides against P. cajani in vitro and in vivo to control zoospore induction as well as infection of zoospores at seedling stage. The fungicides were selected based on their mode of action and targeted impact on the fungus. The in vitro results indicated that Acrobat had the lowest EC50 value (0.17μg/ml). Among fungicide application methods, seed-treatment+soil-drench, soil-drench+foliar-spray and soil-drench are effective in controlling of Phytophthora blight at p<0.001. Combination of seed-treatment+soil-drench and soil-drench+foliar-spray of RidomilGold or Acrobat fungicide on moderately resistant cultivars gives synergistic effect in controlling of Phytophthora blight at seedling stage. This study provides new insight into the control of sporangia formation and zoospore discharge. Overall, this manuscript was well written and the experiments were well done. Some minor points could be improved or explained.
1. In the whole manuscript, the fungicides were referred as “novel”, what does novel mean? Newly developed or not studied for P. cajani?
2. In Abstract, For each fungicide effective concentration (EC50) should be half maximal effective concentration (EC50).
Author Response
The authors thank the reviewers for their valuable inputs which has enhanced the quality of our manuscript. The suggested changes have been duly made to the best of our knowledge.
1. In the whole manuscript, the fungicides were referred as “novel”, what does novel mean? Newly developed or not studied for P. cajani?
Ans: Not studied for P. cajani
2. In Abstract, For each fungicide effective concentration (EC50) should be half maximal effective concentration (EC50).
Ans: As suggested, changed in the Abstract